# Genome-Wide Identification, Evolution, and Expression Characterization of the Pepper (*Capsicum* spp.) *MADS-box* Gene Family

**DOI:** 10.3390/genes13112047

**Published:** 2022-11-06

**Authors:** Zhicheng Gan, Xingxing Wu, Sage Arnaud Missamou Biahomba, Tingting Feng, Xiaoming Lu, Nengbing Hu, Ruining Li, Xianzhong Huang

**Affiliations:** Center for Crop Biotechnology, College of Agriculture, Anhui Science and Technology University, Chuzhou 233100, China

**Keywords:** *Capsicum annuum*, *Capsicum baccatum*, *Capsicum chinense*, MADS-box family, comparative evolution, purifying selection

## Abstract

MADS domain transcription factors play roles throughout the whole lifecycle of plants from seeding to flowering and fruit-bearing. However, systematic research into *MADS*-*box* genes of the economically important vegetable crop pepper (*Capsicum* spp.) is still lacking. We identified 174, 207, and 72 *MADS*-*box* genes from the genomes of *C*. *annuum*, *C*. *baccatum*, and *C*. *chinense*, respectively. These 453 *MADS*-*box* genes were divided into type I (Mα, Mβ, Mγ) and type II (MIKC* and MIKC^C^) based on their phylogenetic relationships. Collinearity analysis identified 144 paralogous genes and 195 orthologous genes in the three Capsicum species, and 70, 114, and 10 *MADS*-*box* genes specific to *C*. *annuum*, *C*. *baccatum*, and *C*. *chinense*, respectively. Comparative genomic analysis highlighted functional differentiation among homologous *MADS*-*box* genes during pepper evolution. Tissue expression analysis revealed three main expression patterns: highly expressed in roots, stems, leaves, and flowers (*CaMADS93*/*CbMADS35*/*CcMADS58*); only expressed in roots; and specifically expressed in flowers (*CaMADS26*/*CbMADS31*/*CcMADS11*). Protein interaction network analysis showed that type II CaMADS mainly interacted with proteins related to flowering pathway and flower organ development. This study provides the basis for an in-depth study of the evolutionary features and biological functions of pepper *MADS*-*box* genes.

## 1. Introduction

*MADS*-*box* genes comprise a large family of genes distributed throughout the plant kingdom and therefore occupy an important position in plant growth and development. The MADS acronym is composed of the initials of four proteins: a yeast transcription factor (MCM1), the *Arabidopsis thaliana* floral homozygote AGAMOUS (AG), an *Antirrhinum majus* floral homozygote (DEFICIENS), and human serum response factor (SRF) [1,2]. All four proteins have a highly conserved region consisting of 56–58 amino acids called the MADS domain [3,4]. Approximately one billion years ago, a duplication event occurred in the common ancestor of *MADS*-*box* genes, resulting in two distinct branches, type I and type II [5]. Type I proteins contain mainly SRF structural domains, and type I *MADS*-*box* genes can be further divided into Mα, Mβ, and Mγ; only a few types I genes have a biological function [6]. Type II genes are divided into MIKC^C^ and MIKC* subtypes based on their structural features [7].

Replication and evolution of the type I *MADS*-*box* genes appear to be faster than those of type II genes, but these observations are based on few studies, mainly on the function of type II *MADS*-*box* gene in flower development [8]. According to the classical model of flower development “ABCDE”: class A genes regulate the formation of sepals; class A and B genes together regulate petals; class B and C genes control the differentiation of stamens; class C and D genes are mainly involved in the formation of ovules; class E genes are involved in the regulation of flower organs during the formation process [9,10,11]. In *A.thaliana*, *APETALA1* (*AP1*) represents a class A gene [12]; *APTALA3* (*AP3*) and *PISTILATA* (*PI*) genes belong to class B [13]; *AG* is a representative gene with class C function [14]; *SEPALLATA* (*SEP*) genes belong to class E [15], including *SEP1*, *SEP2*, *SEP3*, and *SEP4*; and class D genes *SEEDSTICK* (*STK*) and *SHATTERPROOF1* (*SHP1*) [16]. In addition, the functions of some genes regulating flowering time, such as *FLOWING LOCUS C* (*FLC*), *SHORT VIRAL PHASE* (*SVP*), and *SUPPORT OF OVEREXPRESSION OF CONSTANS 1* (*SCO1*), have been confirmed in *A. thaliana*, rice (*Oryza sativa*), and wheat (*Triticum aestivum*) [17,18,19].

*MADS*-*box* genes are involved in many plants’ growth and development processes. *MADS*-*box* genes play important regulatory roles in fruit growth and development, such as the *FOREVER YOUNG FLOWER* (*SlFYFL*) gene, the SEP-type *SlCMB1* gene, and auxin-related *SlIAA9* in tomato (*Solanum lycopersicon*) [20,21,22]; *PaMADS7* of cherry (*Cerasus pseudocerasus*) [23]; *MA*-*MADS5* of banana (*Musa nana*) [24]; and *VEGETATIVE TO PRODUCTIVE TRANSITION 2* (*TRV2*) in wheat (*Triticum aestivum*) [25]. *MADS*-*box* genes are also involved in the plant stress response, such as *Zymoseptoria tritici ZtRlm1*; *AtAGL16*; and *SiMADS51* in *Setaria italica* [26,27,28]. Therefore, the MADS-box family is one of the driving factors in plant diversity and plays an important role in growth and development [29,30].

Pepper (*Capsicum* spp.) originated in Central America and the Andes mountains, growing in tropical and temperate environments [31]. At present, 27 species of *Capsicum* have been identified [32], with five species cultivated long-term: *Capsicum annuum*, *C*. *baccatum*, *C*. *chinese*, *C*. *frutescens*, and *C*. *pubescens* [33]. Evolutionary analysis shows that *C. annuum* differentiated from *C*. *chinense* around 1.14 million years ago, and *C*. *baccatum* differentiated from *C. annuum* and *C*. *chinense* 1.7 million years ago [34]. At present, there is little research on the MADS-box gene family in pepper. A *CaMADS-box* gene is involved positively in low-temperature, salt, and osmotic stress signaling pathways [35]. *CanMADS1* and *CanMADS6* genes are expressed in flower buds and fruits and are highly expressed at 2 days after flowering, suggesting involvement in regulating pepper fruit development [36]. However, the members of the MADS-box gene family in pepper have not been systematically identified or analyzed. In this study, we carried out genome-wide identification of the MADS-box gene family from *C*. *annuum*, *C*. *baccatum*, and *C*. *chinense* genome data [34,37,38]. The objectives of this study are to identify and chacterize the *MADS*-*box* genes from three *Capsicum* species: *C*. *annuum*, *C*. *baccatum*, and *C*. *chinense* using genome-wide survey. As a result, 174, 207, and 72 *MADS*-*box* genes were identified from above three *Capsicum* species, respectively. Collinearity analysis identified 144 paralogous genes and 195 orthologous genes in the three peppers, and *MADS*-*box* genes underwent genome replication events in three pepper species during evolution. Protein interaction network analysis revealed that CaAP1 and CaAG were at the core of the network. These results revealed that type II *MADS*-*box* genes play vital roles in flower organ development of pepper. This study provides a theoretical basis for further study of revealing the functions of *MADS*-*box* genes in peppers and for the molecular breeding of peppers.

## 2. Material and Methods

### 2.1. Plant Materials

The seeds of *C*. *annuum* were collected in Sep 2019 from the Fangyang (32°37′ N and 87°46′ E) of the mid-reach of Huaihe River in Anhui, China. *C*. *annuum* seeds planted in pots containing soil:vermiculite:perlite (2:1:1) and placed in a growth chamber under long-day conditions (16 h^−1^ light/8 h^−1^ dark, 23/20 °C day/light, 150 µmol m^–2^ s^–1^). For tissue expression analysis, roots, stems, leaves were collected at the third true-leaf expanding stage, flower, sepal, petals, stamens, and pistils were harvested at flowering stage. Pepper fruits that grew to 3 cm long were sampled. All samples were immediately snap-frozen with liquid nitrogen and then stored in a −80 °C refrigerator until RNA extraction.

### 2.2. Identification and Naming of MADS-box Family Genes in Three Peppers

MADS-box protein sequences of *A*. *thaliana* were downloaded from the TAIR database (http://www.arabidopsis.org, accessed on 6 November 2021), and genomic data of *Capsicum annuum*, *Capsicum baccatum* and *Capsicum chinense* were downloaded from the Pepper Genome Platform (http://peppergenome.snu.ac.kr/, accessed on 6 November 2021) [34]. Algorithm-based BLASTP was performed using the MADS-box protein sequence of *A*. *thaliana* as the query in the protein databases of *C*. *annuum*, *C*. *baccatum* and *C*. *chinense*, with an *E*-value < 1 × 10^−5^ and other parameters as default values. The obtained candidate protein sequences were compared with Pfam (http://pfam.xfam.org/, accessed on 6 November 2021) database using HMMER (http://www.hmmer.org/, accessed on 6 November 2021). The MADS-box domain based on SRF domain (PF01486) and K domain (PF00319) was used for further comparison and screening, and the parameters were the default values. Thus, the MADS-box gene family members of three species were identified and named according to the order of their position on the chromosome. In order to view the distribution of MADS-box on the chromosomes of three pepper more directly, the online website MG2C (http://mg2c.iask.iN/mg2c_v2.0/, accessed on 6 November 2021) was used to draw the chromosome location map. The theoretical molecular weights and isoelectric points of MADS-box proteins were computed by the ExPASy (https://web.expasy.org/protparam/, accessed on 6 November 2021) proteomics server. The subcellular localization of CaMADS-box, CbMADS-box and CcMADS-box proteins were predicted by the ProtComp 9.0 (http://liNux1.softberry.com/berry.phtml, accessed on 6 November 2021) server.

### 2.3. Phylogenetic Tree Construction, Gene Structure and Protein Motif Analysis

The ClustalW program was used to perform multiple sequence alignments between the MADS-box gene family protein sequences of *C*. *annuum*, *C*. *baccatum*, *C*. *chinense*, and *A*. *thaliana*, with the default parameters [39]. MEGA11 was used to construct Maximum Likelihood phylogenetic trees and analyze the evolutionary relationship of MADS-box gene families among different species [40]. The phylogenetic trees were visualized using the EvolView server (https://www.evolgenius.info/evolview/#/treeview, accessed on 6 November 2021) [41]. Analysis of *MADS*-*box* gene exon-intron distribution based on *C. annuum*, *C. baccatum*, and *C. chinense* genome gff3 files using GSDS (http://gsds.cbi.pku.edu.cn/, accessed on 6 November 2021) visualization server. The conserved motifs of the CaMADS-box, CbMADS-box, and CcMADS-box family were identified using the MEME website (http://meme-suite.org/tools/meme, accessed on 6 November 2021). The motif length range was set to 10–60 amino acid residues, the maximum number of motif discoveries was set to 10 and other parameters were set to default values.

### 2.4. Selective Pressure Analysis

Paralogous and orthologous of *MADS*-*box* genes in *C*. *annuum*, *C*. *baccatum*, and *C*. *chinense* were identified using the Ortho Venn2 online website (https://orthovenn2.bioinfotoolkits.net/home, accessed on 6 November 2021) [42]. DNaSP 6.0 software was used to calculate the non-synonymous substitution rate (*K*a) and the synonymous substitution rate (*K*s) [43], and the selection pressure of replicated gene pairs during evolution was evaluated by calculating the ratio *Ka*/*Ks*. *K*a/*K*s > 1, <1 or =1 represent positive, negative or neutral evolution, respectively [44]. Tbtools [45] was used to visualize the collinearity relationship among gene members of paralogous and orthologous *MADS*-*box* genes in three *Capsicum* species.

### 2.5. Pepper MADS-box Gene Expression Analysis and qPCR Validation

RNA-seq data of *C*. *annuum*, *C*. *baccatum*, and *C*. *chinense* transcriptomic were obtained from the BioProject database (https://www.ncbi.nlm.nih.gov/bioproject, accessed on 6 November 2021), and the transcriptome accession number of the three *Capsicum* species was PRJNA223222, PRJNA308879 and PRJNA331024, respectively. [34]. The fastp [46] and RSEM tools [47] were used to filter and compare sequencing data, and the comparison was achieved using the Bowtie2 tool [48]. Parameters are set to default values. The results were standardized using the fragments per kilobase of transcript per million mapped reads (FPKM) of a gene. After the FPKM value was converted by log2(FPKM + 1), a heat map was created using the TBtools software [45], and the expression of *CaMADS-box*, *CbMADS-box* and *CcMADS-box gen*es in different tissue was analyzed.

RNA from each tissue was extracted using TRIzol kit (Life Technologies, Carlsbad, CA, USA); reverse transcription was performed using HiScript III RT SuperMix for qPCR (Vazyme, Nanjing, China) kit; qPCR analysis was performed using ChamQ Universal SYBR qPCR (Vazyme, Nanjing) reagent. The PCR instrument was an ABI ViiA7 real-time fluorescence quantitative PCR machine (Life Technologies, USA). The Primer 3.0 tools (https://bioinfo.ut.ee/primer3−0.4.0/, accessed on 6 November 2021) were used to design *CaMADS-box* gene-specific amplification primers, using *CaUBI3* as the reference gene [49] (Appendix A). The qPCR primers are listed in Appendix A. All qPCR assays were performed using three independent biological replicates, each with three technical replicates. The PCR conditions consisted of 2 min of initial denaturation at 94 °C followed by 40 cycles of 30 s at 94 °C, 45 s at 62 °C, 30 s at 72 °C, and a final 5 min extension at 72 °C. The PCR conditions and the calculation method of relative gene expression were the same as before [50].

### 2.6. Protein Interaction Network Prediction and Validation

The AraNetV2 tool (http://www.inetbio.org/aranet/, accessed on 6 November 2021) and the STRING (http://string-db.org/cgi, accessed on 6 November 2021) databases were used to identify the orthologous pairs between type II *CaMADS-box* and *AtMADS-box* genes. Then, the predicted protein–protein interaction network was displayed through Cytoscape software [51]. The yeast two-hybrid (Y2H) assays were performed in accordance with a method described previously [52,53]. The full-length CDSs of *CaAG* and *CaAP1* were separately cloned into a pGBKT7-GAL4 vector (Clontech, Mountain View, CA, USA) as bait plasmids. The full-length CDSs of *CaSEP3* and *CaSVP* were separately cloned into pGADT7 as the prey constructs (primers in Appendix A). The resulting prey and bait constructs were cotransformed into the yeast strain AH109 as per the description of BD library construction and screening kit (Clontech, USA). The transformants were grown on SD/-Leu/-Trp medium and then transferred to SD/-Leu/-Trp/-His-Ade medium with 3 mM 3-AT to detect the interaction.

## 3. Results 

### 3.1. Genome-Wide Identification and Characterization of the MADS-box Transcription Factor Gene Family from Capsicum L.

We identified 174, 207, and 72 MADS-box gene family members from three *Capsicum* species: *C*. *annuum*, *C*. *baccatum*, and *C*. *chinense*, respectively (Appendix A), by referring to the amino acid sequences of *A*. *thaliana* MADS-box proteins, local BLAST comparison, and screening using the Pfam (http://pfam.xfam.org/, accessed on 6 November 2021) website. The number of genes in *C*. *annuum* and *C*. *baccatum* was more than that in *C*. *chinense*. All 453 MADS-box proteins possessed conserved SRF and K domains; the genes encoding these proteins were named *CaMADS1*-*CaMADS174*, *CbMADS1*-*CbMADS207*, and *CcMADS1*-*CcMADS72*, respectively. Analysis of physicochemical properties of MADS-box proteins in the three pepper types found that the amino acid (aa) lengths of CaMADS-box, CbMADS-box, and CcMADS-box proteins were 59–661, 49–660, and 78–1100 aa, respectively (Appendix A). Their molecular weights were 6813–74021.51, 5691.74–74516.8 and 9052.62–125758.3 KDa, respectively. Isoelectric points ranged from 4.51 to 10.94, 4.89 to 10.23 and 4.46 to 10, respectively (Appendix A). Prediction of subcellular localization showed that most MADS-box proteins were located in the nucleus, with a few located outside the cellular (Appendix A).

### 3.2. Categorization, Structural Classification, and Structure of MADS-box Genes in Pepper

A phylogenetic tree was reconstructed using 105 *AtMADS-box* and 453 pepper *MADS*-*box* genes to further study the phylogenetic relationships of *MADS*-*box* genes. Following the classification and structure of the *A. thaliana* MADS-box family, we reconstructed two separate phylogenetic trees of type I (Appendix A) and type II (Figure 1) genes, respectively. The results showed that type I MADS-box genes were divided into three subfamilies (Mα, Mβ, and Mγ). There were 99, 6, and 16 members of Mα, Mβ, and Mγ, respectively, in *C*. *annuum*; 134, 8, and 7 members, respectively, in *C*. *baccatum*; and 34, 3, and 6 members, respectively, in *C*. *chinense* (Appendix A). The Mα subfamily in *C*. *baccatum* was considerably expanded, while the Mβ subfamily in *C*. *chinense* was substantially contracted. Type II MADS-box genes were further divided into 10 subfamilies: SEP, AGL16, AP1, STK/AG, AGL12, SCO1, SVP, PI/AP3, FLC/MAF, and MIKC*. The “ABCDE” genes of flower development, such as *SEP*, *AP1*, *AG*, *PI*, *AP3*, and *STK*, were amplified among the type II genes of the three pepper species (Figure 1, Appendix A).

Conserved motifs of MADS-box family proteins were analyzed using the online website MEME. A total of 10 motifs with a length of 15–41 amino acids were predicted (Appendix A), and their distribution trend was conserved within every subfamily (Appendix A). Motifs 1 and 3 were common to most MADS-box transcription factors, but motif 6 was unique to type II protein members. Motif 5 was also a unique domain for MIKC-type protein members.

Most *MADS*-*box* genes belonging to the same subfamily exhibited the same pattern of gene structure, but there were great differences among different members (Appendix A). Most type I genes was composed of one exon, but *CbMADS28*, *CbMADS81*, *CbMADS124*, and *CcMADS47* had two exons. Meanwhile, type II genes contained multiple introns and exons. Compared with type II genes, type I genes had suffered intron loss. Differences in exon and intron structure between type I and type II genes may be one of the reasons for the increase in MADS-box gene family members during evolution.

### 3.3. Phylogenetic Relationships of MADS-box Genes in Pepper

To investigate homologous *MADS*-*box* genes in pepper and possible gene duplication in each pepper species, we next identified the evolutionary relationships of *MADS*-*box* genes in the three *capsicum* species. There were 39 groups of orthologous genes, accounting for 22.9% (40/174) of *MADS*-*box* genes in *C*. *annuum*, 19.3% (40/207) of *MADS*-*box* genes in *C*. *baccatum,* and 55.5% (40/72) of *MADS*-*box* genes in *C. chinense* (Figure 2). This indicated that some *MADS*-*box* genes were preserved during evolution of *C*. *annuum* and *C*. *baccatum*, while *MADS*-*box* genes were highly conserved during the evolution of *C. chinense*. In addition to 40 groups of orthologous genes shared by the three pepper species (Appendix A), 193 *MADS*-*box* homologs were found between any two pepper species (Appendix A). There were 85 pairs of orthologous genes between *C*. *baccatum* and *C*. *annuum*, 61 pairs of orthologous genes between *C*. *annuum* and *C*. *chinense*, and 47 pairs of orthologous genes between *C*. *baccatum* and *C*. *chinense* (Appendix A). Furthermore, *C*. *annuum*, *C*. *baccatum,* and *C*. *chinense* possessed 70, 117, and 10 unique *MADS*-*box* genes, respectively (Figure 2). There were 140 pairs of paralogous *MADS*-*box* genes, of which 47 pairs were tandem repeats (Appendix A). Among these duplicated genes, some displayed have one-to-many relationships, such as *CaMADS40*, which was the tandem repeat gene of both *CaMADS39* and *CaMADS41*. This was reflected in *C*. *baccatum*, with tandem repeat genes of *CbMADS20* including *CbMADS18*, *CbMADS17*, *CbMADS9*, *CbMADS8,* and *CbMADS191*. However, there were no one-to-many situations in *C*. *chinense*. This indicates that the MADS-box gene family of pepper has an obvious gene replication phenomenon, which explains why the number of *MADS*-*box* genes in *C*. *annuum* and *C*. *baccatum* is more than that in *C*. *chinense*.

To further explore the evolutionary mechanisms of *MADS*-*box* genes in pepper, we constructed collinear circos of the three *Capsicum* species based on orthologous genes and tandem repeat genes (Figure 3). The results revealed that *MADS*-*box* genes are distributed on every chromosome, mainly located at the terminus of each chromosome arm. Orthologous genes in the three *Capsicum* species were very close on the chromosome, anchored in a highly conserved collinearity block. Based on the genomic information available so far, we found 39 genes not localized to chromosomes in *C*. *annuum*, 69 genes not localized to chromosomes in *C. baccatum*, and 1 gene not localized to a chromosome in *C*. *chinense*. This phenomenon may also result from the error generated during chromosome assembly or the poor quality of assembly [34].

In conclusion, the results revealed that the MADS-box transcription factor family of pepper is somewhat conserved. The 10 pairs of genes found in *C*. *annuum* formed linear relationships between pairs (Figure 3), indicating that duplication occurred between *MADS*-*box* genes. We calculated the selection pressure of paralogous as well as orthologous genes of the MADS-box gene family in *Capsicum* spp. pepper. The results showed that among the paralogous homologs (Figure 4A), *K*a/*K*s < 1 for all paralogous genes in *C. annuum*, while in *C*. *baccatum*, there were 32 pairs of paralogs with *K*a/*K*s > 1 (Appendix A). This indicates that *C. annuum* was subject to strong purifying selection during its evolution, whereas *C*. *baccatum* was susceptible to environmental changes. However, no *K*a/*K*s values for paralogous genes were detected in *C*. *chinense*. Among the orthologs in the three *Capsicum* species, *Capsicum annuum* (Ca), *C*. *baccatum* (Cb), and *C*. *chinense* (Cc) (Figure 4B), the mean *Ka*/*Ks* values of the orthologs of Ca-Cb, Ca-Cc, and Cb-Cc were 0.6055, 0.6003, and 0.5952, respectively, with Ca-Cb having the largest mean value, implying that the *MADS*-*box* homolog of Ca-Cb was subject to greater purifying selection.

### 3.4. Comparative Evolutionary Relationships of Type II MADS-box Genes in Three Capsicum Speciess

To study the contraction and expansion of type II MADS-box family members during evolution, the phylogenetic relationships among MIKC *MADS*-*box* genes in the three pepper species were explored using collinearity analysis. The results revealed 33 pairs of colinear genes between *C*. *annuum* and *C*. *baccatum*, and 18 pairs of colinear genes between *C*. *baccatum* and *C*. *chinense* (Figure 5, Appendix A). Most type II genes were located at both ends of chromosomes, such as *CaMADS8* on chromosome 1 and *CbMADS80* on chromosome 8. In addition, the homologous genes on chromosomes 1, 2, 11, and 12 of *C*. *annuum* were distributed on chromosomes 4 and 5 of *C*. *baccatum*; the type II homologous genes on chromosomes 1 and 2 of *C*. *annuum* were located on chromosomes 6 and 3 of *C*. *chinense*. In *C*. *baccatum*, the type II homologous genes of chromosomes 1, 5, and 8 were distributed on chromosomes 1, 2, and 6 of *C*. *chinense*. There was no type II MADS-box gene on chromosome 9 in the three *Capsicum* species, which may be related to interchromosome 9 translocations [34]. In summary, most of the type II MADS-box genes in the three peppers showed conserved collinearity among chromosomal regions, but there was also deviation in duplicated gene pairs.

### 3.5. Expression Characteristics of MADS-box Genes in Different Pepper Tissues

We next analyzed the expression profiles of the *MADS*-*box* genes in root, stem, leaf, and flower tissues using RNA-seq data for the three peppers. The results showed that the expression of *MADS*-*box* genes in 73 groups of orthologous genes was considerably different among the three pepper species in the four tissues (Figure 6). Comprehensive analysis revealed that their expression patterns could mainly divide the genes into three categories: (1) genes expressed in all four tissues, such as *CaMADS93*/*CbMADS35*/*CcMADS58*, *CaMADS82*/*CbMADS198*/*CcMADS4*, and *CaMADS116*/*CbMADS134*/*CcMADS66*, indicating that they are widely involved in the growth and development of pepper; (2) genes with high expression during flower development, such as *CaMADS26*/*CbMADS31*/*CcMADS11*, *CaMADS30*/*CbMADS33*/*CcMADS14*, and *CaMADS74*/*CbMADS83*/*CcMADS41*, belonging to type II, suggesting that they play important roles in flower development; and (3) genes with high expression in roots, such as *CaMADS9*/*CbMADS81*/*CcMADS2* and *CaMADS68*/*CbMADS74*/*CcMADS36*, indicating that these genes may be involved in root development and some physiological and biochemical processes in underground plant part. In addition to the three distinct expression patterns, most of the orthologous genes showed the same expression trend in the three pepper species, but some homologous genes displayed different expressions in the same tissue. For example, *CcMADS69* was not expressed in any tissues, while its homologous gene *CaMADS127* was highly expressed not only in flowers, but also in stems, indicating that orthologous genes in pepper may have gained or lost functions in the process of evolution.

To further observe the expression of the type II MADS-box genes in different tissues of pepper, an expression heat map of 39 *CaMADS-box* genes in root, stem, leaf, and flower tissues was drawn (Appendix A). Six MADS-box genes, *CaMADS74*, *CaMADS30*, *CaMADS61*, *CaMADS26*, *CaMADS105*, and *CaMADS63*, were selected for analyzing expression levels in root, stem, leaf, flower, and fruit tissues using qPCR. The results revealed that the six *MADS*-*box* genes were differentially expressed in different tissues of pepper (Appendix A), but they were highly expressed in flowers, which was consistent with the results of RNA-seq data (Figure 6). Both *CaMADS26* and *CaMADS30* were highly expressed in flowers, moderately expressed in fruits, and almost not expressed in other tissues. Both *CaMADS61* and *CaMADS74* were highly expressed in flowers, with little or no expression in other tissues, but *CaMADS74* was weakly expressed in roots. Expression of *CaMADS63* was the highest in flowers, followed by fruits and leaves, and low or trace expression was found in other tissues. However, the expression of *CaMADS105* was higher in fruits than in flowers, and low or no expression was found in other tissues.

We next further analyzed the expression profiles of these six type II genes using qPCR in sepal, petal, stamen, and pistil tissues (Appendix A). In the sepal, *CaMADS61* expression was the highest, followed by *CaMADS105* and *CaMADS74*. *CaMADS30* was moderately expressed, while *CaMADS63* and *CaMADS26* were weakly expressed. *CaMADS61* was highly expressed in petals, while the expression levels of *CaMADS105* and *CaMADS30* were relatively low. For other genes, there was little or no expression in petals. In stamen tissue, *CaMADS63*, *CaMADS26,* and *CaMADS74* showed slightly expressed; *CaMADS105* and *CaMADS30* were moderately expressed; and *CaMADS61* was highly expressed. *CaMADS74* was the highest expression in the pistil, followed by *CaMADS105*. *CaMADS30*, *CaMADS26*, and *CaMADS63* were slightly expressed, but *CaMADS74* was not expressed in the pistil.

### 3.6. Interaction Network of Type II CaMADS Proteins

To better understand the biological functions of type II MADS-box genes in pepper, we next predicted the interaction network of CaMADS proteins. The results revealed that only 17 type II CaMADS-box members interacted with each other (Figure 7). The interacting proteins were mainly flowering pathway proteins and flower organ development proteins, of which AP1, AG, SEP3, AGL20, and AGL21 were at the core of the network. *AtAP1* regulates the transition of inflorescence meristem and the morphological development of flower organs [54]. *AtAG* controls the stamens and carpels and inhibits the expression of *AtAP1* [55]. *AtSEP3* belongs to the class D gene, which involved in the process of flower development and activates the function of *AtAG* [56]. Moreover, Y2H assay confirmed that CaAG (CaMADS26) interacted with CaSVP (CaMADS50), and CaSEP3 (CaMADS105) could interact with CaAP1 (CaMADS24) and CaAG, respectively (Figure 8).

Connection between nodes varies with the combined_score value (representing the reliability of the predicted interaction between the two proteins, ranging between 0 and 1), thickening with an increase in score value. The size of nodes increases with the number of proteins interacting with node proteins.

Triangles represent a 10-fold dilution, with T, L, H, and A representing Tryptophan, Leucine, Histidine, and Adenine, respectively.

## 4. Discussion

Gene duplication often accompanies plant evolution and is an important reason for the expansion of gene families [57]. The MADS-box family is one of the largest transcription factor families and plays an important role in growth and development, and signal transduction [4,58]. With the development of sequencing technology, MADS-box gene family members have been identified in a variety of plants in varying numbers, such as 107 *MADS*-*box* gene members in *A. thaliana* [7], 83 in *Camellia sinensis* [59], 44 in *Nelumbo nucifera* [60], 44 in *Erigeron breviscapus* [61], 131 in *Solanum lycopersicum* [62], 54 in *Morella rubra* [63], 42 *Phyllostachys heterocycle* [64], 80 in *Triticum aestivum* [65], 54 in *Ziziphus jujuba* [66], 144 in *Raphanus sativus* [67], 82 in *Lactuca sativa* [68], 160 in *Brassica rapa* [69], 78 in *Callicarpa americana* [70], and 108 in *Chrysanthemum nankingense* [71]. These studies indicate that *MADS*-*box* genes have undergone obvious amplification and contraction, and the number and distribution in different subfamilies are also different. We study identified 174, 207, and 72 *MADS*-*box* genes from *C*. *annuum*, *C*. *baccatum,* and *C*. *chinense*, respectively (Figure 1; Appendix A), in line with this trend. Moreover, the number of MADS-box family genes in *C*. *baccatum* was more than that in *C*. *annuum* and *C*. *chinense*, which may be due to the expansion of the *C*. *baccatum* genome caused by the amplification of retrotransposons [34]. The number of *MADS*-*box* genes of type I and MIKC subfamilies in *C. annuum* and *C. baccatum* was more than that of the model plant *A*. *thaliana* and the related species tomato. These results indicate that the *MADS*-*box* genes in *C. annuum* and *C. baccatum* have significantly expanded, but strangely, the number of genes identified in *C*. *chinense* was lower than that in *C. annuum* and *C*. *baccatum*, suggesting that a large number of genes have been lost during evolution. Among the MADS-box family members, those belonging to the same subfamily possessed similar motif composition and gene structure, but there was a unique motif composition and gene structure between type I and type II genes. The *MADS*-*box* genes of type I, including Mα, Mβ, and Mγ, generally contained no or few exons (Appendix A) and may have lost multiple introns during the diversification of the MADS-box family. In addition, the distribution of introns in pepper *MADS*-*box* genes was also different. MIKC–type genes had more introns than those of type I, which are also found in A. thaliana, tomato, and rice [72], indicating that evolution between species is conserved. However, some genes of the same subfamily showed different intron and exon arrangements, indicating the complexity of gene structure evolution, which needs further study. The same conserved motifs in the same subfamily (Appendix A) suggest that these motifs play an important role in gene functional specificity. Analyses of gene structure and conserved motifs provide clues for the expansion and contraction of the MADS-box gene family in pepper.

In eukaryotes, gene replication plays an important role in amplifying the number of transcription factor families and genomic complexity [57,62]. Previous studies confirmed that *C*. *annuum* diverged from *C*. *chinense* 1.14 million years ago and *C*. *baccatum* diverged from *C*. *chinense* and *C*. *annuum* 1.7 million years ago [34]. Our study revealed only 47 groups of orthologous genes among three pepper species, while other orthologous genes were lost to a certain extent (Figure 2). We also identified 144 groups of paralogous genes and 195 groups of orthologous genes (Figure 3, Appendix A). However, some homologous genes were lost in the three species of pepper. These results further demonstrate both obvious contraction and expansion trends that the different subfamily members of the MADS-box gene family during the process of pepper evolution.

Genome sequencing revealed a dynamic rearrangement of chromosomes 3, 5, and 9 in *C*. *baccatum*, namely, translocation [34]. *MADS*-*box* genes at these loci were also changed, such as *CbMADS35* on chromosome 3 of *C*. *baccatum*, and its homologous gene *CcMADS58* on chromosome 9. In addition, some homologous genes were also located on different chromosomes. Moreover, most of the orthologous genes with *K*a/*K*s greater than 1 displayed positive selection and may show positive changes in function under the influence of the environment (Figure 4).

Most *MADS*-*box* genes were differentially expressed in different tissues of pepper (Figure 6 and Appendix A), indicating their functional diversity in different tissues. Some *MADS*-*box* genes showed tissue-specific expression, such as *CaMADS9*/*CbMADS81*/*CcMADS2* and *CaMADS66*/*CbMADS75*/*CcMADS35*, which were mainly specifically expressed in roots, and are important candidate genes for further functional analysis. Some *MADS*-*box* genes were highly expressed in fruits (Appendix A), such as *CaMADS30*, *CaMASDS61*, *CaMADS63*, and *CaMADS105*, suggesting important roles in controlling fruit development. Several studies have proved that *MADS*-*box* genes play an important role in the morphogenesis and growth of roots and fruits [73,74]. *CaMADS105* were expressed in sepal, petal, stamen, and pistil (Appendix A), suggesting that they play vital roles in all stages of flower development. MIKC *MADS*-*box* genes play a central role in plant development [75].

In this study, we predicted the possible key genes in pepper flower organ development through phylogenetic relationships, such as class A genes (*CcMADS13*/*CaMADS31*/*CbMADS34*), class B genes (*CcMADS41*/*CaMADS74*/*CbMADS83*), class C genes (*CcMADS11*/*CaMADS26*/*CbMADS31*), class D genes (*CcMADS8*/*CaMADS1*7/*CbMADS27*), and class E genes (*CcMADS14*/*CaMADS30*/*CbMADS33*, *CcMADS61*/*CaMADS101*/*CbMADS114*). Protein interaction network analysis revealed (Figure 7) that type II MADS-box proteins were mainly proteins related to flowering regulation and flower organ development. Y2H assay confirmed that CaAG interacted with CaSEP3 and CaSVP, and CaSEP3 interacted with CaAP1 and CaAG, suggesting that CaAG, CaSEP3, CaSVP, and CaAP1 play important roles in flowering, and the formations of sepal, petal, carpel, and stamen. *AtAG* is the only C functional gene in *Arabidopsis*, which interacts genetically with other allotypic genes to identify floral organs [76]. In summary, we comprehensively identified the *MADS*-*box* gene members of the three *Capsicum* species and analyzed their structural characteristics and evolutionary rules. The MIKC *MADS*-*box* genes identified in this study should be candidate genes for pepper breeding and improvement.

## Figures and Tables

**Figure 1 genes-13-02047-f001:**
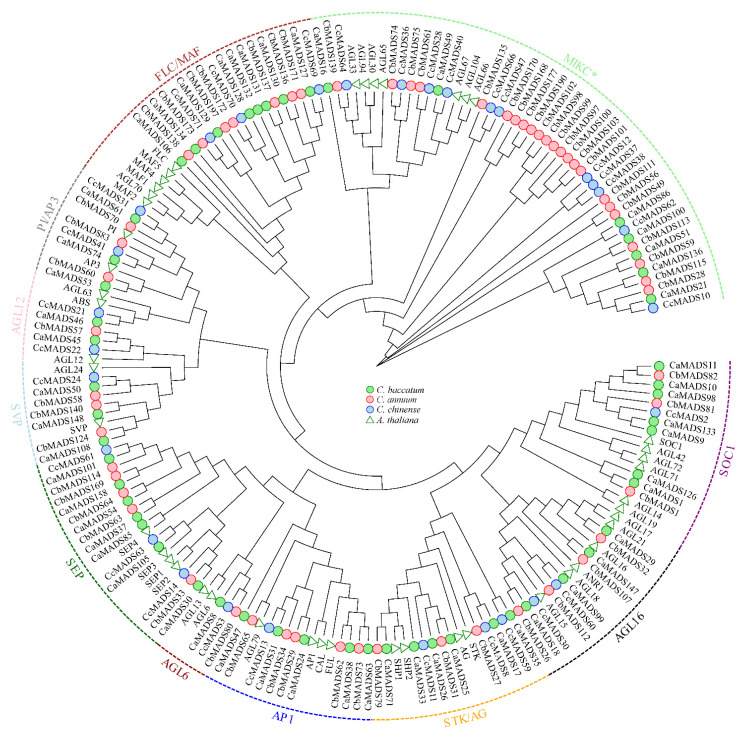
Phylogenetic tree of type II *MADS*-*box* genes in *Arabidopsis thaliana*, *Capsicum annuum*, *C*. *baccatum*, and *C*. *chinense*.

**Figure 2 genes-13-02047-f002:**
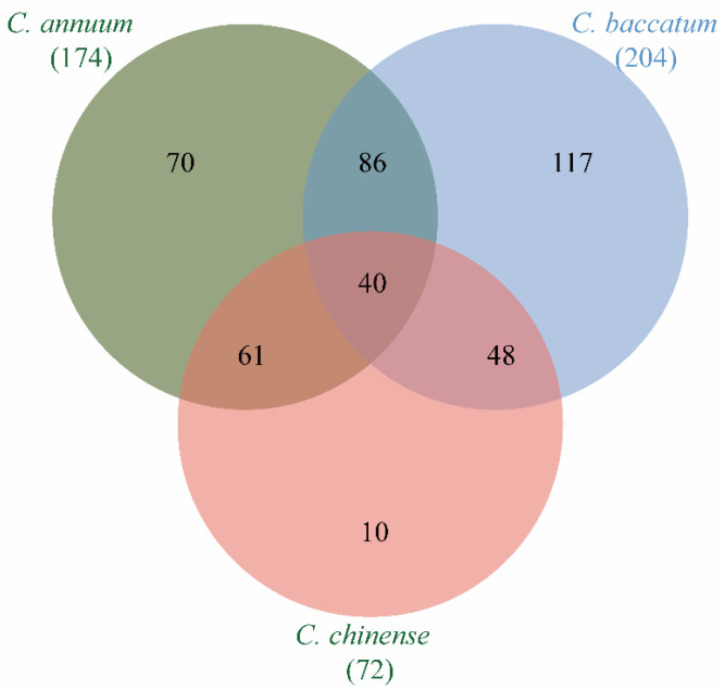
Number of *MADS*-*box* orthologs in *C. annuum*, *C. baccatum*, and *C*. *chinense*.

**Figure 3 genes-13-02047-f003:**
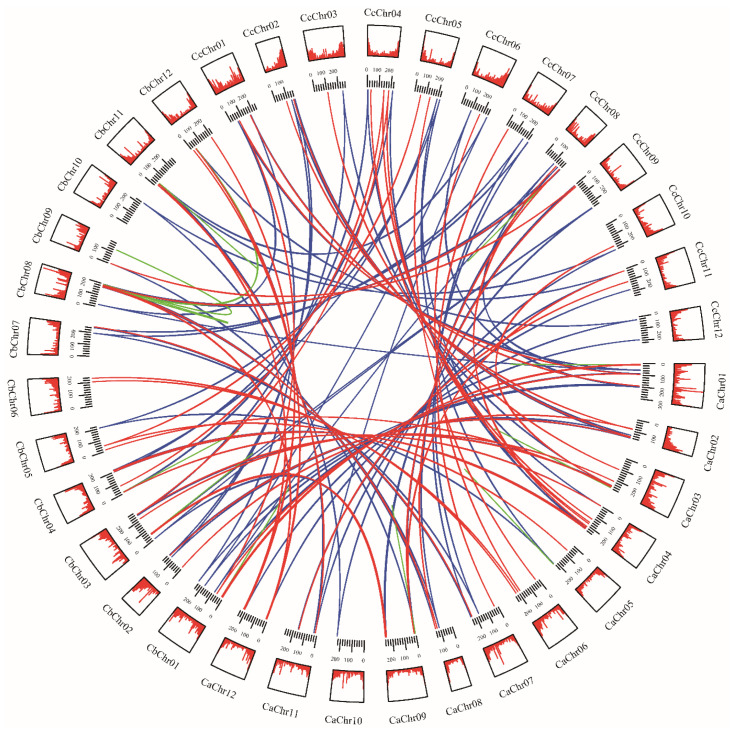
Homologous *MADS*-*box* gene pairs in *Capsicum annuum* (Ca), *C*. *baccatum* (Cb), and *C*. *chinense* (Cc). Tracks from outside to inside are chromosomes numbers, gene density of the chromosome, and homologous gene pairs among the three *Capsicum* species. Blue lines connect homologous gene pairs that exist in three *Capsicum* species; red lines connect homologous gene pairs in two species; green lines connect paralogous genes.

**Figure 4 genes-13-02047-f004:**
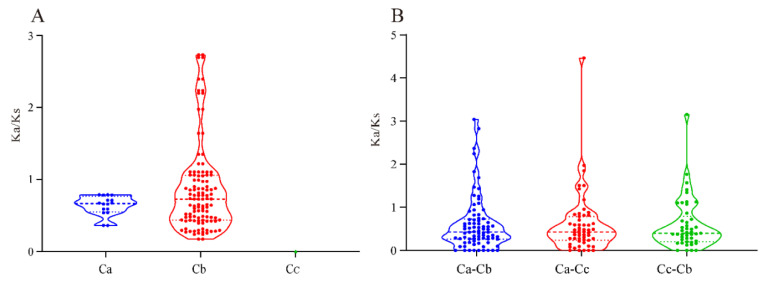
Selection pressure statistics of paralogous (**A**) and orthologous (**B**) *MADS*-*box* genes in pepper. Ca, *C. annuum*; Cb, *C. baccatum*; Cc, *C. chinense*. The ‘dots’ reflect the maximum and minimum *K*a/*K*s scores.

**Figure 5 genes-13-02047-f005:**
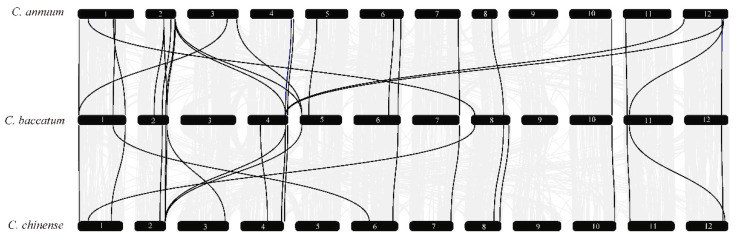
Collinearity of type II MADS-box genes in *C. annuum*, *C. baccatum*, and *C. chinense*.

**Figure 6 genes-13-02047-f006:**
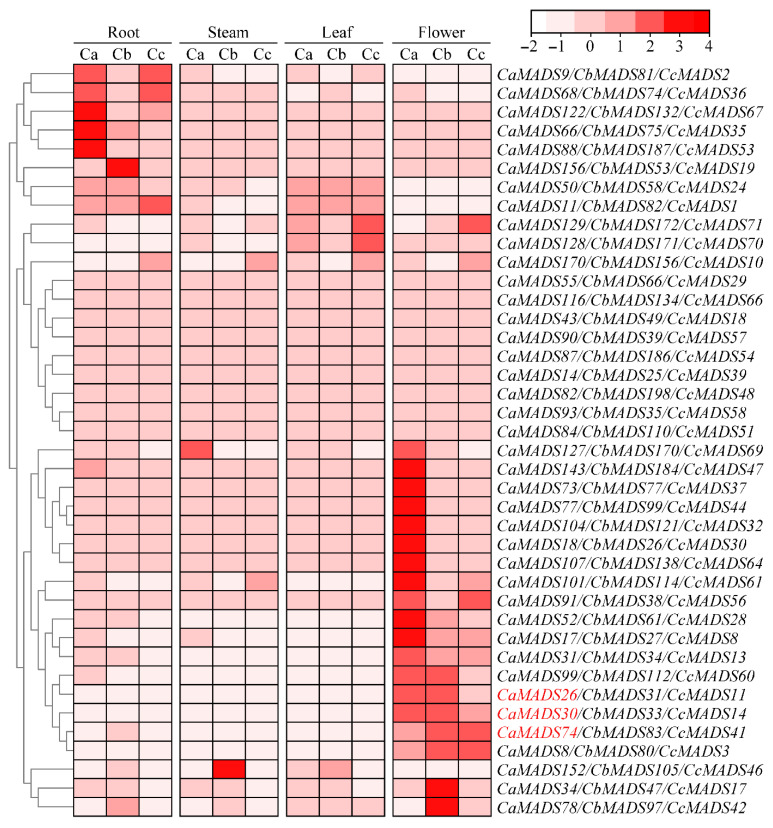
Expression profiles of *MADS*-*box* genes in different tissues from *Capsicum annuum* (Ca), *C*. *baccatum* (Cb), and *C*. *chinense* (Cc). Color bar indicates the variation range of log_10_(FPKM + 1) values of *MADS*-*box* genes in different tissues. Expression of genes marked in red was verified by qPCR.

**Figure 7 genes-13-02047-f007:**
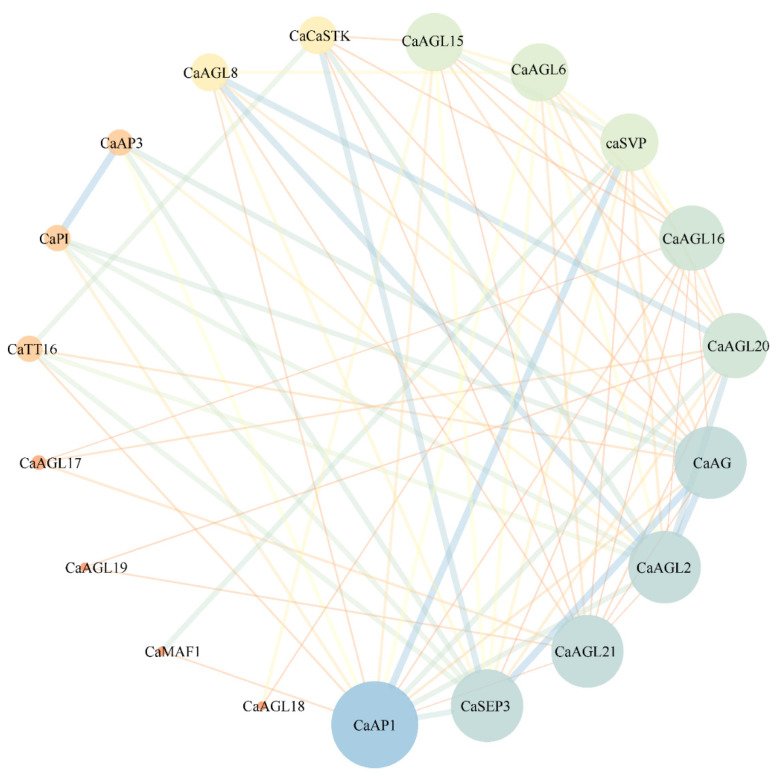
Type II CaMADS-box Protein Interaction Network Diagram.

**Figure 8 genes-13-02047-f008:**
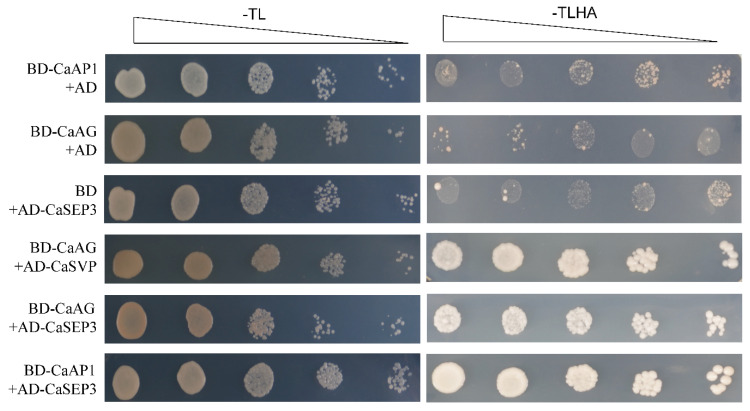
Yeast two-hybrid assays of interactions among CaAP1, CaAG, CaSVP, and CaSEP3 proteins.

## Data Availability

MADS-box gene members of *A*. *thaliana* were downloaded from the TAIR database (http://www.arabidopsis.org, accessed on 6 November 2021). Genomic data of *C. annuum*, *C. baccatum* and *C. chinense* were downloaded from the Pepper Genome Platform (http://peppergenome.snu.ac.kr/, accessed on 6 November 2021). *C. annuum*, *C. baccatum* and *C. chinense* transcriptome sequencing data were downloaded from the BioProject database (https://www.ncbi.nlm.nih.gov/bioproject, accessed on 3 April 2021).

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
