# Peer review of "Genome-Wide Identification, Evolution, and Expression Characterization of the Pepper (Capsicum spp.) MADS-box Gene Family"

_genes, 2022, doi:10.3390/genes13112047_

Round 1

Reviewer 1 Report

Dear authors

This manuscript regarding Genome-Wide Characterization and Expression he pepper MADS-box gene family. This manuscript is well performed with clear objectives and discussion; the manuscript results corroborate with previous studies that the MADS-box gene has a significant role in development; especially in flowers and fruit development. Thus, I would like to recommend this manuscript for publication after minor revision.

A few of my comments are below for the advancement of the manuscript:

Why authors used MEGA6.0? Authors must use a higher version of MEGA like MEGA10 or MEGA11. Do authors use all 453 MADS-box genes from Capsicum spp. for Phylogenetic tree construction? If not (due to the variable length of the gene; Neighbor-Joining methods not suitable), the author may use the Maximum Likelihood method for Phylogenetic tree construction.

Heading no. for the Discussion section should be 4.

I would also like to see a few references to recent studies. It would seem that the authors have not included a few important studies. Some recent references may be included in the appropriate section (introduction or discussion) such as:

Duan W, Song X, Liu T, Huang Z, Ren J, Hou X, Li Y. Genome-wide analysis of the MADS-box gene family in Brassica rapa (Chinese cabbage). Molecular genetics and genomics. 2015 Feb;290(1):239-55.

Ning K, Han Y, Chen Z, Luo C, Wang S, Zhang W, Li L, Zhang X, Fan S, Wang Q. Genomewide analysis of MADSbox family genes during flower development in lettuce. Plant, Cell & Environment. 2019 Jun;42(6):1868-81.

Alhindi T, Al-Abdallat AM. Genome-Wide Identification and Analysis of the MADS-Box Gene Family in American Beautyberry (Callicarpa americana). Plants. 2021; 10(9):1805.

Won SY, Jung JA, Kim JS. Genome-wide analysis of the MADS-box gene family in Chrysanthemum. Computational Biology and Chemistry. 2021 Feb 1;90:107424

Author Response

Thank you for your comments and suggestions on our mauscripts very much, which are very useful for improving our manuscript. We have carefully revised our manuscript accorrding to your suggestions.

We provided a point-by-point response to your comments. Please see the attachment.

Reviewer 2 Report

Comments and Suggestions for Authors

The following are some general points and scientific queries that need to be addressed in the

Introduction: What is the hypothesis, objectives, and novelty of the work; please clearly indicate at the end of the introduction.

Materials and Methods are not clearly described. Before starting the design of any scientific research, the scientific researcher must choose the scientific method that fits the subject of his research study.

The materials and methods section should be more cogent; it should have more details. I would recommend explaining in detail. Therefore, we advise the researcher to follow the steps of writing the scientific research in detail and clearly describe it.

Results: Describe the major findings of your study in the opening sentence. Writing results need improvement. I would recommend checking all web addresses

Discussion: they should discuss the findings briefly and relate them to their study.

References: The reference section must be consistent and prepared based on the journal.

Author Response

Thank you for your comments and suggestions on our manuscript very much. These comments greatly helped us improve our manuscript and provided important guidance for future research. We have carefully revised our manuscript accorrding to your suggestions.

We provided a point-by-point response to your comments. Please see the attachment.

Reviewer 3 Report

Dear Prof. Dr. Editor of Genes,

I write you regarding Manuscript Number: genes-1997410 entitled "Genome-wide identification, evolution, and expression characterization of the pepper MADS-box gene family" which was submitted to Genes Journal.

In this manuscript, the authors studied the

 Genome-wide identification, evolution, and expression characterization of the pepper MADS-box gene family. This work was well done, and the methods of data collection were appropriate.

I have gone through this work. My decision is accepted with minor revisions for this work. The reason for that is as follows:

The manuscript deals with "Genome-wide identification, evolution, and expression characterization of the pepper MADS-box gene family.

First: Title: It should change to the following:

1)   Genome-wide identification, evolution, and expression characterization of the pepper (Capsicum annuum L.) MADS-box gene family

 Or

Genome-wide identification, evolution, and expression characterization of the pepper (Capsicum spp.) MADS-box gene family

Second Abstract, keywords and Introduction:

2) has some minor corrections as in the attached file.

Third: The objectives of the study

3) is ok.

Fourth: Materials and Methods

4) has some minor corrections as in the attached file.

Results and discussion

5)  has some minor corrections as in the attached file.

References

6) the references please check and correct some minor corrections as in the attached file.

Thank you for suggesting me as a reviewer for this paper.

with best regards

Author Response

Thank you for your positive comments and suggestions on our manuscript very much. These comments greatly helped us improve our manuscript. We have carefully revised our manuscript accorrding to your suggestions.

We summarized a point-by-point response to your comments. Please see the attachment.

Round 2

Reviewer 1 Report

Dear Authors,

Revised manuscripts have been much improved. I am satisfied with the response to the comments. Thus, I would like to recommend it for acceptance in its current form.

Author Response

Thank you for your reviewing our manuscript again. We have checked all the text carefully according to all referees’ suggestions and amended all questions.

Reviewer 2 Report

Dear Authors,

I am informing you that your manuscript can be accepted for publication as revised. 

Thank you for giving us the opportunity to consider your work.

Author Response

Thank you for your recognition of our paper. And then, we have checked all the text carefully again according to all referees’ suggestions and amended all questions.

Reviewer 3 Report

Dear Prof. Dr Editor-in-Chief of Genes,

I write you regarding Manuscript Number: genes-1997410-peer-review-v2 entitled "Genome-wide identification, evolution, and expression characterization of the pepper MADS-box gene family" which was submitted to Genes Journal.

My decision is accepted for this work after reference corrections. There are fetal mistakes in references i.e., the scientific name must be written in corrected forms. The species must write in small letters this is not corrected as notes in the previous review.

Thank you for suggesting me as a reviewer for this paper.

with best regards

Author Response

Thank you for your careful reviews very much! We apologize for the questions displayed in the section of ′References′. This time, we carefully checked all the references one by one. The scientific names in the references have been corrected which were marked with the "track changes" function.